# The Use of Translational Modelling and Simulation to Develop Immunomodulatory Therapy as an Adjunct to Antibiotic Treatment in the Context of Pneumonia

**DOI:** 10.3390/pharmaceutics13050601

**Published:** 2021-04-22

**Authors:** Robin Michelet, Moreno Ursino, Sandrine Boulet, Sebastian Franck, Fiordiligie Casilag, Mara Baldry, Jens Rolff, Madelé van Dyk, Sebastian G. Wicha, Jean-Claude Sirard, Emmanuelle Comets, Sarah Zohar, Charlotte Kloft

**Affiliations:** 1Department of Clinical Pharmacy & Biochemistry, Institute of Pharmacy, Freie Universitaet Berlin, 12169 Berlin, Germany; sebastian.franck@fu-berlin.de (S.F.); charlotte.kloft@fu-berlin.de (C.K.); 2Unit of Clinical Epidemiology, Assistance Publique-Hôpitaux de Paris, CHU Robert Debré, Université de Paris, Sorbonne Paris-Cité, Inserm U1123 and CIC-EC 1426, F-75019 Paris, France; moreno.ursino@inserm.fr; 3INSERM, Centre de Recherche des Cordeliers, Sorbonne Université, Université de Paris, F-75006 Paris, France; sandrine.boulet@crc.jussieu.fr (S.B.); sarah.zohar@inserm.fr (S.Z.); 4HeKA, Inria, F-75006 Paris, France; 5Department of Clinical Pharmacy, Institute of Pharmacy, University of Hamburg, 20146 Hamburg, Germany; sebastian.wicha@uni-hamburg.de; 6CNRS, Inserm, CHU Lille, Institute Pasteur de Lille, U1019-UMR9017-CIIL-Centre for Infection and Immunity of Lille, Université de Lille, F-59000 Lille, France; fcasilag@gmail.com (F.C.); mara.baldry@inserm.fr (M.B.); jean-claude.sirard@inserm.fr (J.-C.S.); 7Department of Evolutionary Biology, Institute of Biology, Freie Universitaet Berlin, 14195 Berlin, Germany; jens.rolff@fu-berlin.de; 8Flinders Centre for Innovation in Cancer, College of Medicine and Public Health, Flinders University, Adelaide 5042, Australia; madele.vandyk@flinders.edu.au; 9INSERM, University Rennes-1, CIC 1414, F-35000 Rennes, France; emmanuelle.comets@inserm.fr; 10INSERM, IAME, Université de Paris, F-75006 Paris, France

**Keywords:** pharmacometrics, translational modelling, anti-infective therapy, antibacterial resistance, innate immunity, pneumonia, dose estimation, PK/PD, Bayesian inference, extrapolation

## Abstract

The treatment of respiratory tract infections is threatened by the emergence of bacterial resistance. Immunomodulatory drugs, which enhance airway innate immune defenses, may improve therapeutic outcome. In this concept paper, we aim to highlight the utility of pharmacometrics and Bayesian inference in the development of immunomodulatory therapeutic agents as an adjunct to antibiotics in the context of pneumonia. For this, two case studies of translational modelling and simulation frameworks are introduced for these types of drugs up to clinical use. First, we evaluate the pharmacokinetic/pharmacodynamic relationship of an experimental combination of amoxicillin and a TLR4 agonist, monophosphoryl lipid A, by developing a pharmacometric model accounting for interaction and potential translation to humans. Capitalizing on this knowledge and associating clinical trial extrapolation and statistical modelling approaches, we then investigate the TLR5 agonist flagellin. The resulting workflow combines expert and prior knowledge on the compound with the in vitro and in vivo data generated during exploratory studies in order to construct high-dimensional models considering the pharmacokinetics and pharmacodynamics of the compound. This workflow can be used to refine preclinical experiments, estimate the best doses for human studies, and create an adaptive knowledge-based design for the next phases of clinical development.

## 1. Introduction

Pneumonia is caused by fungal, viral, or bacterial infections of the lung, with the latter causing the majority of pneumonia infections [1]. Particularly, *Streptococcus pneumonia* accounts for two-thirds of bacterial lung infections [2]. Pneumonia is the most common lower respiratory tract infection (LRTIs) and the leading infectious cause of death globally [1,3]. Despite this disease being largely preventable, it has claimed around 2.6 million lives every year for the past decade, making it the fourth leading cause of death overall. While in the past decade the deaths per year have decreased for children (under 5 years old), the opposite is true for the elderly (over 65 years old) [1,3]. Community- and hospital-acquired bacterial pneumonia (CAP and HAP, respectively) are principal causes of morbidity, quality-adjusted life year loss, and mortality in the entire population, with annual incidence rates between 0.1–1% (CAP, [4]) and up to 20% (HAP, [5]), with even higher incidences associated with ventilator use (VAP, [5]). These bacterial infections are typically treated with antibiotics, but their effectiveness is declining. Indeed, due to inappropriate use or dosing of antibiotics, spread of resistant strains, and insufficient awareness, antimicrobial resistance (AMR) is rapidly rising [6,7,8,9,10,11]. The World Health Organization (WHO) calls antibiotic resistance “one of the biggest threats to global health, food security, and development today” and estimates that bacterial infections due to AMR will outcompete any cause of death by 2050 [12]. In parallel, new antibiotics are barely investigated or further developed until market approval. It is thus crucial to develop alternative strategies to improve antibacterial treatment [13,14,15]. To address this, the WHO has declared the optimized use of antimicrobial medicines as one of the key strategic objectives as part of tackling this global antibiotic resistance challenge [12]. Additionally, AMR may be further complicated by the COVID-19 pandemic due to the increase in broad-spectrum antimicrobial drugs used for COVID-19-related presumptive bacterial infections, despite the low bacterial co-infections actually observed in COVID-19 patients. For example, it was observed that 72% of COVID-19-positive hospitalized patients received antibiotics when only 8% showed bacterial co-infections [16]. This inappropriate or unnecessary use of antimicrobials largely occurred due to the lack of decision support tools for the clinical management of COVID-19 and absence of rapid diagnostic tools [17].

Alternative strategies to tackle bacterial infections have steadily been on the rise in the last decade [18,19,20] and include phage therapy [21], antimicrobial peptides [22,23], bacterial monoclonal antibodies [19], combination therapy [24] and immunomodulatory compounds [25]. While most mentioned alternatives are targeting the pathogen, the latter capitalizes on the defense systems already present, i.e., the innate immunity of the host. In this way, many of the challenges regarding AMR are circumvented, as the pathogen is not targeted directly and is thus not selectively pressured, even though indirect pressure might still exist [26]. However, overly stimulating the immune system can result in severe adverse events in the patients, such as systemic inflammatory response, which can lead to the potentially lethal cytokine release syndrome and cytokine storm syndrome [27]. Therefore, a good understanding of the interaction between immunomodulatory compounds and the host is of utmost importance for a successful clinical application of this strategy. Furthermore, as these compounds are often large molecules that need to be delivered to the target-site, the development from preclinical to clinical is challenging, and no gold standard approaches exist to date [19,20,25,26,28]. Here, we aim to provide a framework that can be applied to tackle this challenging translation.

When developing a compound from the preclinical stage to clinical application, the choice of a safe starting dose for early phase first-in-man clinical trial needs to be considered. This dose should be low enough to avoid a high rate of (serious) adverse events while being efficacious enough and not too low to avoid time-consuming dose escalation studies [29]. In order to extrapolate the dose from preclinical species to humans, several methods exist. These methods are based on multiple assumptions and, in fields other than oncology, can be classified in four classes. These classes are (1) the use of the no observable adverse effect level (NOAEL) from preclinical toxicology studies multiplied by a safety factor [30]; (2) the pharmacokinetically guided approach (PGA) that uses systemic exposure rather than dose for the extrapolation from animal to man [31]; (3) the use of NOAEL information from similar drugs (e.g., same chemical class) that may be already at clinical stage; and (4) the comparative approach, where more methods are used and then the results are critically compared [32]. Once the starting dose is chosen, the full dose panel of the first-in-man clinical trial must be defined and, usually, algorithmic approaches such as the modified Fibonacci dose escalation are used [33]. Along with the well-known algorithm designs for phase I dose escalation clinical trials (i.e., the A + B designs), Bayesian model-based designs (such as the continual reassessment method [34] and the Bayesian logistic regression model [35]) have been drawing an increasing amount of attention.

In phase I studies, even if dose-finding and pharmacokinetic/pharmacodynamic (PK/PD) analyses are carried out in the same trial, they are often conducted and reported independently in different sections in publications reporting trial results [36]. However, a recent simulation study showed how integrating PK information into the dose escalation design could enrich the knowledge of the dose–toxicity relationship, thus facilitating better dose recommendations for subsequent trials [37]. Recently, new designs that incorporate PK measurements into the dose escalation process [38] or PK/PD in the final dose–response curve estimation [39] have been proposed. Therefore, in order to maximize the probability of clinical success of alternative therapies such as immunomodulation, all available knowledge regarding PK, PD, and toxicology should be modelled and applied as early in the drug research and development process as possible. In this way, information collected early on can be used to design further experiments and can be carried forward throughout the clinical development, strengthening the understanding of the compound and ultimately allowing appropriate clinical use.

Pharmacometric approaches, which use mathematical models to quantitatively describe data from different sources, including stochastic models to describe variability between individuals, animals, or experiments, lend themselves perfectly for the consolidation of knowledge to efficiently translate a compound from the preclinical to the clinical setting [40,41,42,43,44,45]. They can be applied for this translation in complex settings, such as inhalation administration [46,47,48], oncology [49], antibody–drug conjugates [50], and pediatrics [51]. In order to incorporate the uncertainty regarding all these different sources of data in the translational process, pharmacometric models can be combined with the Bayesian approaches [52]. Bayesian statistics provides a mathematical framework to incorporate prior knowledge (quantified as prior distributions), and to update them to posterior knowledge (expressed as a posterior probability distribution), via the likelihood. A recent study describing a Bayesian meta-analytic approach applied by Zheng et al. suggests how to use preclinical data to inform the design and prior distributions [53]. Indeed, the Bayesian approach has also been recognized as a powerful and flexible method in the pharmacometric field [54,55] and has been applied in sequential preclinical trials [56] to build informative prior distributions in human PK analysis using preclinical information [57] or using information from adult clinical studies to design pediatric trials [58].

In this concept paper, we report on how translational modelling and simulation approaches can effectively be used in the context of the clinical development of immunomodulatory drugs as adjunct to antibiotic therapy. For this, two case studies are presented: the combination of antibiotics with either the Toll-like receptor (TLR) 4-agonist monophosphoryl lipid A (ABIMMUNE) or the TLR5 agonist flagellin (FAIR). Indeed, the TLRs are attractive targets for stimulation of the innate immune system as they are conserved across mammals to recognize bacterial threats by binding with different components of the bacterial envelope, cell wall, or flagella [26,59]. Binding of one of these compounds to these receptors leads to a signaling cascade, which, consequently, leads to the production of pro-inflammatory cytokines, chemokines, and other compounds with antimicrobial properties [59]. Here, we describe how derivatives of these bacterial compounds can be used as immunomodulatory drugs and how innovative pharmacometric approaches can aid in the translation of these compounds to clinical use. We focus on the challenges and lessons learned in these case studies, identifying which data are necessary for a successful translation and how these data are best analyzed and applied. To conclude, we present an outlook on how the developed modelling and simulation framework could be applied for future immunomodulatory drug development programs in particular, and drug development in general.

## 2. The ABIMMUNE Project

Two projects aiming to develop immunomodulatory drugs as adjunct to antibiotic drugs in the context of pneumonia are described in this work. In the first one, ABIMMUNE, the lipopolysaccharide (LPS)-derived compound monophosphoryl lipid A (MPLA) was used in combination with amoxicillin (AMX) in order to treat mice infected with Streptococcus pneumoniae. This project is described in detail elsewhere [60,61], and the extensive pharmacometric analysis applied on the data can be found in a companion paper elsewhere in this issue [62].

An extensive in vivo dataset was generated, investigating multiple levels of murine data to comprehensively gain information about the efficacy of the combination of AMX and MPLA. A murine infection model was established with immunocompetent Swiss and Balb/cJ mice being infected intranasally with *S. pneumoniae* during anesthesia. Then, 12 h post infection, treatment was initiated with a single dose of AMX via oral gavage or MPLA intraperitoneally. In total, four study groups were investigated: untreated mice, treatment with AMX, MPLA, or the combination of both drugs. While a single dose tier was defined for MPLA, multiple dose tiers of AMX were investigated. Several markers were assessed with typically one sample per individual mouse in independently performed studies: AMX serum concentrations over time, bacterial numbers in lung and spleen over time, survival, change in body weight, mRNA expression of markers of the immune system, cytokine serum concentrations, histological scores of immune-system-related markers, and cell recruitment of immune-system-related cells to the lung. To quantitatively analyze the investigated combination of AMX and MPLA, pharmacometric analysis of the generated data was performed. Using the collected PK data of AMX and the PD data in terms of bacterial numbers in the lung and spleen and survival, a pharmacometric PK/PD model was developed in a sequential modelling approach. First, a PK submodel and a bacterial disease submodel were linked via an effect compartment submodel and finally extended to a bacterial disease and treatment model. Additionally, the developed PK/PD model was linked to survival in a time-to-event (TTE) analysis. The preclinically obtained results were further investigated to evaluate the validity of potential translational approaches with the aim of facilitating translation of the disease and treatment model into a clinical setting by using, for example, allometric scaling [63] and physiological considerations.

The combination of AMX and MPLA displayed higher efficacy than monotherapy of respective drugs in terms of lower bacterial burden and higher survival. The proposed pharmacometric bacterial disease and treatment model revealed mechanistic and quantitative insights. In addition to the determined PK interaction between AMX and MPLA, the stimulating effects of the immune system by MPLA additively reduced bacterial burden and increased murine survival. Whereas AMX showed an expected short-term effect, MPLA was able to stimulate the immune system over a longer time. This immunostimulatory effect was supported by the pharmacometric modelling approach as well as generated in vivo data representing the effects of the immune system in terms of, e.g., cytokines. In addition, the applied surge function described a high hazard for murine death already within a short time frame of approximately 3 days.

Aiming to translate the preclinical results of the comprehensive pharmacometric analysis into a clinical setting, PK parameters of the PK submodel were translated based on basic allometric scaling techniques and compared to values reported in literature. The extrapolated values were higher and did not accurately match reported human parameters but were at least plausible and in the same magnitude, which is to be expected from basic allometric extrapolation approaches. For a subsequent translation of the PD data in the lung towards humans, an already published human PK submodel for AMX was chosen as the basis [64]. In this case, certain assumptions and simplifications were made: Bacteria-specific parameters (e.g., bacterial growth and natural bacterial elimination) and PD parameters of the disease and treatment model (such as drug-specific parameters) were not scaled in contrast to the initial bacterial number in the lung and the transfer of AMX from serum to the lung. Primarily, the influence of MPLA on bacterial elimination effects was considered not to be different in humans. Here, a single low dose of AMX would theoretically be sufficient for bactericidal killing of a susceptible strain. However, due to large differences in the sensitivity of humans and mice to MPLA [65], this extrapolation remains difficult, especially for resistant strains, and further clinical development was thus halted.

## 3. The FAIR Project

The second project, FAIR, builds on and deepens the learnings from ABIMMUNE and aims to exploit the immunomodulatory compound flagellin, derived from bacterial flagellae. In this project, the compound is being developed up until phase 1 human clinical trials, amalgamating knowledge gained from expert opinion and in vitro and preclinical experiments along the way. By incorporating a pharmacometric approach early on in the development process along with a continuous learning modelling under Bayesian inference, FAIR can capitalize on more knowledge than ABIMMUNE and thus increases the probability of success of translation to the clinical setting.

As a proof-of-concept, it has been shown that flagellin of which the hypervariable region is removed can be used to activate TLR5 without triggering an inflammation reaction [66]. This was further applied in a murine infection model, and evidence for therapeutic synergy between antibiotics and flagellin in the context of pneumonia was obtained [67,68,69]. Based on these data, FAIR continues to generate PK, PD, and toxicology data in in vitro systems and preclinical species in order to develop flagellin towards clinical use as an add-on to the standard of care with antibiotics [70]. In order to combine the different sources of data obtained throughout the project, it was necessary to set up a comprehensive modelling and simulation framework. Based upon the ABIMMUNE pharmacometric analysis on the one hand and a dose-finding statistical modelling approach on the other hand, a decision to incorporate information as early as possible in this framework seemed the most appropriate. Indeed, in FAIR, prior knowledge from the literature (such as expert opinion and results from other clinical studies on the same or similar molecules) is combined with the stepwise newly available in vitro and in vivo data generated by the consortium to construct, in an iterative way, advanced mathematical models characterizing the PK/PD of flagellin. The translational aspect of collecting information (“learn”) and using it for the next step (“predict” and “confirm”) in an iterative way, as well as having the clinical trial subjects’ benefiting from all of the available information at the inclusion time, will allow for a better use of available knowledge and the possibility to update/correct any action in real time. Indeed, at each step, our approach will optimize the next in vitro or in vivo experimentation. By doing so, not only will all the data be incorporated in the full model framework approach but they will also make use of the best knowledge gathered until that point in time. The different PK processes (e.g., absorption, distribution, and elimination) specific for large protein biologics in the context of nebulization are taken into account in the various respiratory and body compartments.

The final goal is to propose the best design for human studies (including phase I) within a clinical trial framework. This implies the use of all available information gathered during the FAIR project, as well as expert data, under a Bayesian framework in order to better design prospective clinical trials. Indeed, by using existing information associated with an optimal dose allocation process during the early phase evaluation of a drug, a better estimation of the dose–response relationship can be achieved [37]. Moreover, once the trial is started, information gathered during the trial including primary and secondary outcomes will be modelled for a better understanding of the dose–response relationship over time. Furthermore, accounting for possible differences in dose–response relationships from the different data sources seems conceptually straightforward and allows the creation of higher dimension models close to the reality [58].

Ultimately, we propose a novel methodology for improving dose finding in early phase clinical trials by incorporating data from (i) (pre)clinical trials realized; (ii) external information, such as expert opinion, patients’ acceptability, and results from other (pre-)clinical studies on the same or similar molecules; (iii) the PK; and (iv) the PD. The use of such integrative design will give a more comprehensive picture of drug safety and the drug mechanism during the dose-finding trial, enabling improved estimation of the best dose level for further evaluation whilst ensuring the sample size for early phase trials is kept to a minimum [71,72].

To the best of our knowledge, this full Bayesian PK/PD dose-finding framework would be the first of its kind and would drastically expand upon the typical independent working of trial statisticians and pharmacometricians during drug development. In this way, we will increase accuracy, optimize the dose estimation process, and eventually bring the most appropriate dose and schedule to phases II and III. The further development and validation of this framework might be used as a blueprint for other drug development programs and could aid in the current high failure rate of clinical trials due to inappropriate dose–response estimation.

## 4. Discussion

### 4.1. Lessons Learned from ABIMMUNE and FAIR

The aim of this concept paper was to use two case studies to elucidate the use of translational modelling and simulation approaches in early drug development, specifically in the case of immunomodulatory drugs in the context of bacterial pneumonia.

In the first project, ABIMMUNE, pharmacometric modelling was used to quantify a PK and a PD interaction between the TLR4 agonist MPLA and the antibiotic AMX. Indeed, by using powerful hierarchical modelling approaches, sparsely sampled murine trials could be combined into one dataset and exploited in full. In this way, the magnitude of the bacterial killing effect of the immunomodulator or antibiotic alone, as well as that of the combination, could be quantified. Furthermore, the pharmacometric PK/PD model was linked to a TTE model, which could quantify the effects of treatment in the sense of survival. The translation of these results towards humans is limited, however, as large differences in immune response to MPLA exist between humans and mice. Furthermore, no immunocompromised mice were studies as a negative control (no immunostimulation), and the study design did not fully consider modelling of the data afterwards. A PK translation of the results was feasible, but for a PD translation, adaptations to this study design would be necessary, using, e.g., immunocompromised or TLR4-deficient mice, and quantifying immune response, PK and PD in the same mice for at least some time points. Furthermore, quantification of MPLA concentrations or the use of different dose levels was lacking, which made a complete characterization of the immunomodulatory effect impossible.

In FAIR, many of the aforementioned limitations are taken as lessons learned and thus overcome. A richer study design involving different preclinical species (both immunocompetent and TLR5-deficient) and human-derived in vitro systems are continuously updated using a modelling and simulation framework. To date, this integrative framework has maximized the use of all prior knowledge together with the first iteration of experiments in order to allow a first dose estimation for use in a toxicology trial. Moreover, a Bayesian framework was proposed, where the different inputs (e.g., results from previous analyses, literature data, and expert elicitation) are weighted via prior distributions, allowing one to preserve all prior knowledge and to update it.

Furthermore, flagellin will be quantified in preclinical species, allowing better characterization of immunomodulatory PK apart from the administered antibiotic, which, in turn, may allow for more accurate PK/PD models and thus higher probability of successful translation towards humans.

A caveat for a successful application of this paradigm is good interdisciplinary communication. As the modelling and simulation framework is dependent on data generated by project partners and, at the same time, provides design input to these studies, a mutual understanding of the importance of this clear communication is paramount. Indeed, apart from the scientific complexities arising from this innovative framework, ensuring a good flow of information between the different sources is one of the main challenges one has to consider when applying a translational modelling and simulation approach in a drug development project.

### 4.2. A Translational Modelling and Simulation Framework for Development of Immunomodulatory Drugs

Based on the two case studies described in this concept paper, we propose a model-informed platform, described in Figure 1, which takes into account the specificity of immunomodulatory drugs and can be used as a blueprint for drug development in this area. The driving force of the framework is the well-known “learn–predict–confirm” paradigm [73], where iteratively more complex models are used to weigh, model, and apply data as they become available. This paradigm should be applied to the standard of care that is considered, i.e., the PK/PD of the antibiotic that is to be combined with the immunomodulatory drug, and to the new (immunomodulatory) drug under development. Based on the built-in Bayesian approaches, the consistency of input data with extrapolated results can be assessed continuously, which allows early-on detection of, e.g., model misspecification. At the beginning of a drug development project, a small amount of information with high uncertainty is available, allowing only for simple pharmacometric models combined with non-informative priors in a Bayesian inference framework. As these models are used to design further experiments, confidence in model structure increases as data become available and priors are updated using posterior distributions of previous analyses and/or external data. As the project continues, a knowledge database for the compound is constructed, and a model repository of increasingly more complex models is created. For example, with more data available, basic allometric scaling approaches can be complemented by mechanistic models, such as physiologically based pharmacokinetic models. Bayesian posterior and/or predictive probabilities then aid in applying these pharmacometric models to obtain starting dose levels for human studies. This continues into the clinical phase, where all generated knowledge is used to propose a first-in-man dose, which still persists within the framework and is thus also uncertain and can be updated as more data become available. Furthermore, models describing the emergence of AMR as a function of (antibiotic) exposure and infection duration (influenced by the immunomodulator) could be added to this framework to quantify the impact of a proposed new combination therapy on the global AMR dynamics [74]. Indeed, by modulating the immune response, the so-called mutant selection window can be narrowed, and linking the immunomodulator model to a model describing the PK/PD of antibiotics and a bacterial growth model with multiple populations, the impact on resistance emergence could be quantified [75,76].

### 4.3. Application in the Current Drug Discovery and Development Landscape

Currently, the process of drug discovery and development is still long and expensive, often taking more than ten years and billions of dollars from first lead candidate to market authorization [77]. The application of modelling and simulation approaches to this process, i.e., model informed drug discovery and development (MID_3_), has been proposed for higher efficiency and probability of success and is currently recommended by the FDA [78]. Although the application of this framework to small molecules is starting off in, e.g., oncology [79] and anti-infectives [80,81,82], there are only few examples that apply such a framework to large molecules, mostly in the case of monoclonal antibodies [83,84]. Given the many challenges but also large opportunities in the development of immunomodulatory drugs, it is crucial to optimally use all available information as early on and efficiently as possible. Furthermore, immunomodulatory drugs are often following a “hit-and-run” mechanism of action, implying that in an early stage, pharmacokinetic information is of lesser importance than dose–response relationships. Innovative experimental in vitro systems such as reconstituted human tissues [85] can provide the first information regarding these relationships at an early stage in the drug development process, and later on this knowledge can be augmented by preclinical experiments. Using a flexible modelling and simulation platform then allows pharmacokinetic information to be incorporated at a later stage, once information about the target site and systemic exposure becomes available. Our proposed translational modelling and simulation platform, which goes beyond standard MID_3_ paradigms by incorporating PK/PD models within a Bayesian framework, could be used to speed up the development of immunomodulatory compounds, crosslinking the preclinical, clinical, and in silico development in a multidisciplinary information-driven approach.

## 5. Conclusions

Translational modelling and simulation approaches can be of tremendous support in the development of immunomodulatory drugs. When combined with a Bayesian inference framework, different sources of data can be amalgamated while taking into consideration the uncertainty of each source. In this way, a maximally informed path towards the best human dose is created, and the probability of clinical success is increased. We proposed a blueprint for this framework based on real-world projects, which can be used for future drug research and development programs. More informative priors for drug research and development programs could be derived from this framework by applying it to a number of immunostimulatory drugs to extract general drug-independent principles. Additionally, this framework could be applied to the study of bacterial resistance evolution as the immunostimulatory drug changes the selective environment, which could result in lower probability of resistance evolution.

In order to ensure successful application of pharmacometric approaches in drug research and development, they should be taken into account as early on as possible, preferably before the start of the project, and multiple preclinical species with specific characteristics, such as immunocompromised or TLR-deficient animals, should be investigated to optimally inform the development of the model. Furthermore, continuous communication between data analysis and data generation groups is paramount to ensure a steady flow of information and ultimately pave the way towards clinical development. Lastly, while the approach suggested here was based on the development of immunomodulatory drugs in the context of pneumonia, it can easily be generalized to other compounds and therapeutic areas.

## Figures and Tables

**Figure 1 pharmaceutics-13-00601-f001:**
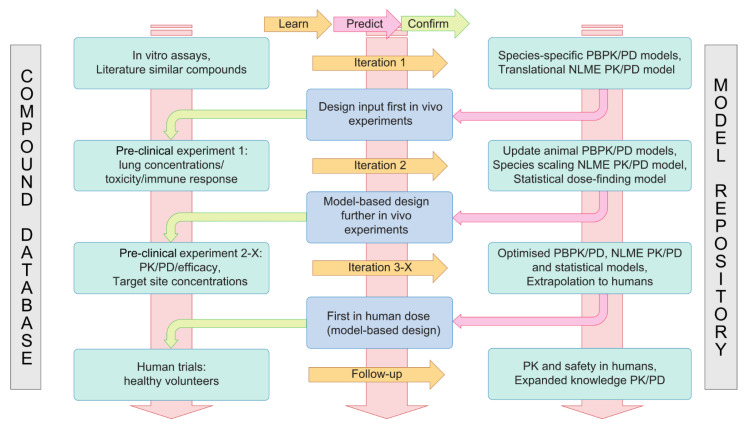
The developed translational modelling and simulation platform for integration, exploitation, and modelling of an immunomodulatory compound’s PK/PD. Starting from the literature and pre-existing datasets, the platform enables the design of appropriate preclinical experiments and novel data to implement the model in order to perform dose finding for clinical trials. This process can be iterated multiple times to predict the PK/PD and doses to be used in phase I trials and beyond. PD: pharmacodynamics; PK: pharmacokinetics; NLME PK/PD: nonlinear mixed effects PK/PD; PBPK/PD: physiologically based PK/PD.

## Data Availability

Data can be provided by the authors when possible upon request.

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
