# Peer review of "The Use of Translational Modelling and Simulation to Develop Immunomodulatory Therapy as an Adjunct to Antibiotic Treatment in the Context of Pneumonia"

_pharmaceutics, 2021, doi:10.3390/pharmaceutics13050601_

Round 1
Reviewer 1 Report
In this conceptual paper, the authors proposed to use pharmacometric approaches to help the development of immunomodulatory drugs as adjunct to antibiotic therapy. It is expected that the modelling and simulation framework can be beneficial for translating these drugs into clinical use and make drug development more efficient. Overall, the manuscript is well written and clearly structured, easily accessible with broad readership.
Here are some questions for the authors to consider:
- The abbreviations shall be clearly explained at the first place. (e.g. Line 135 MPLA, Line 141 AMX)
- The authors were aware of using the basic allometric scaling techniques, which may not be able to extrapolate accurately and match reported human parameters. It is recommended that a more mechanistic interspecies scaling method should be considered for the allometric scaling. In vitro in vivo extrapolation (IVIVE) might be included as the scaling methods to minimize the bias from interspecies differences in metabolism on human clearance prediction.
- Since immunomodulatory drugs were used to reduce the emergence of bacterial resistance, it is suggest that the emergence of resistance should be incorporated to increase the predictive ability of the model.
- Before these PKPD models can be fully utilized, the authors might consider to confirm that predictions based on the models are in agreement with the previously determined PK and PD characteristics of the antibacterial agents.
Reviewer 2 Report
The purpose of this manuscript was not clear to the reviewer. It would be really helpful to guide what is the goal of this manuscript? Although submitted as an Original research manuscript, the reviewer was not able to understand how the structure of this manuscript is applicable for an original research article.
Apart from that the reviewer also has few questions regarding the overall concept being discussed in this manuscript.
- What is ModSim platform? Is it a platform developed by the authors or applied in the discussed trials (ABIMMUNE and FAIR)? More information might be very helpful.
- The concluding sentence in the ABIMMUNE trial section was "However, due to large differences in the sensitivity of humans and mice to MPLA [63] this extrapolation remains difficult and clinical development was halted". The authors clearly touched upon an important point regarding the extrapolation of PD from preclinical to clinical. Can the authors elaborate on how their platform or method be able to avoid this for other similar studies?
- The authors presented a good-to-have or great-to-have information based approach for the that is generally well accepted in drug discovery and development. More details is required to appreciate the novelty for the proposed approach.
Round 2
Reviewer 2 Report
Thank you for the detailed response to the reviewer's comments.
The proposed platform presented by author in Figure 1 is as described my the authors, "Based on the two case studies described in this concept paper, we propose a blueprint to apply to other (immuno- modulatory) drug development platforms". Also, the current manuscript is describing development of immunomodulatory therapy "as an adjunct to antibiotic treatment in the context of pneumonia".
How is the platform described in Figure 1 different than a generalized model-informed drug development platform that can be applied for any other therapeutic area? How this platform can be specifically applied for the development of immunomodulatory therapy as an adjunct to antibiotic treatment in the context of pneumonia?
In the reviewer's opinion authors can be more specific about the end points and objectives of the learn-predict-confirm cycle that is relevant to the therapeutic area of the current manuscript.
Round 3
Reviewer 2 Report
Thank you for the detailed response.